# Mutations observed in somatic evolution reveal underlying gene mechanisms

Michael W. J. Hall [1], David Shorthouse [2], Rachel Alcraft [3], Philip H. Jones [1,4] & Benjamin A. Hall [2✉]

Highly sensitive DNA sequencing techniques have allowed the discovery of large numbers of somatic mutations in normal tissues. Some mutations confer a competitive advantage over wild-type cells, generating expanding clones that spread through the tissue. Competition between mutant clones leads to selection. This process can be considered a large scale, in vivo screen for mutations increasing cell fitness. It follows that somatic missense mutations may offer new insights into the relationship between protein structure, function and cell fitness. We present a flexible statistical method for exploring the selection of structural features in data sets of somatic mutants. We show how this approach can evidence selection of specific structural features in key drivers in aged tissues. Finally, we show how drivers may be classified as fitness-enhancing and fitness-suppressing through different patterns of mutation enrichment. This method offers a route to understanding the mechanism of protein function through in vivo mutant selection.

[1] Wellcome Sanger Institute, Hinxton CB10 1SA, UK. [2] Department of Medical Physics and Biomedical Engineering, Malet Place Engineering Building, University College London, Gower Street, London WC1E 6BT, UK. [3] Advanced Research Computing, University College London, London, UK. [4] Department of Oncology, University of Cambridge, Cambridge CB2 0XZ, UK. ✉email: b.hall@ucl.ac.uk

Over the last decade, DNA sequencing has enabled the detection of vast numbers of somatic mutations in normal tissues[1–6]. Many mutations have no effect on cell behaviour, but may generate large mutant clones by chance, through 'neutral drift'[7,8]. However, a subset of protein altering mutations change cellular properties, increasing mutant cell 'fitness' above that of wild type cells. Such mutants may either disrupt the function of protein encoded by one allele of a haploinsufficient gene or generate a gain of function mutant[8]. Mutants of this type drive clonal expansions and are much more likely than neutral mutations to generate mutant clones. In sequencing studies, these mutants are identified as being positively selected, meaning there is a statistical excess of protein altering over synonymous mutations in the gene[9]. Conversely, protein altering mutations that decrease the fitness of the mutant cell will be outcompeted by wild type cells, depleted from the tissue, and negatively selected[10].

The normal tissue sequencing studies reported to date have revealed large numbers of protein altering mutations in positively selected mutant genes that, in some cases, significantly exceed the number of mutations in the same gene identified in cancers of the same tissue[8]. If mutations in positively selected genes have functional impact, they may drive clonal expansion in vivo. It follows that normal tissue sequencing data provides a potentially valuable resource for the study of protein structure/function relationships.

To mine this data, it is essential to classify the functional impact of mutants. This may be done by aggregating all functional impacts using a single score or categorising mutations into broad categories, such as missense or nonsense mutations

(Supplementary Note 1). However, mutations can have diverse impacts - a missense mutation altering the active site may activate a protein while a missense mutation that destabilises the protein core may inactivate it. To associate particular changes in protein function with changes to cell fitness it is essential to test for selection of individual types of functional changes. This objective is complicated by the fact that strong or widespread selection of a particular subset of mutations can obscure the signal from weaker selection in the same region (Fig. 1). Here, we develop a statistical technique that can peel back layers of selection to reveal weaker, but biologically important selection underneath (Fig. 1).

We demonstrate the power of this method using datasets of mutant clones found through DNA sequencing of normal human oesophageal epithelium[1] and skin[6]. These studies collected samples of normal, non-cancerous tissues from individuals of different ages. Due to the large available sample size from these studies and the known functional impact of the frequently occurring mutations, we use *NOTCH1* to introduce and validate the method. We also examine mutations in *FBXW7* to explore mutational patterns and generate hypotheses linking protein function to cell fitness. Between them, these two genes show the versatility and wide applicability of the method. We conclude by showing that proteins which enhance or suppress cell fitness display different patterns of missense mutations.

## Results

**Patterns of selection in *NOTCH1*.** *NOTCH1* is a strong driver of clonal expansion in normal skin and oesophageal epithelium[1,6].

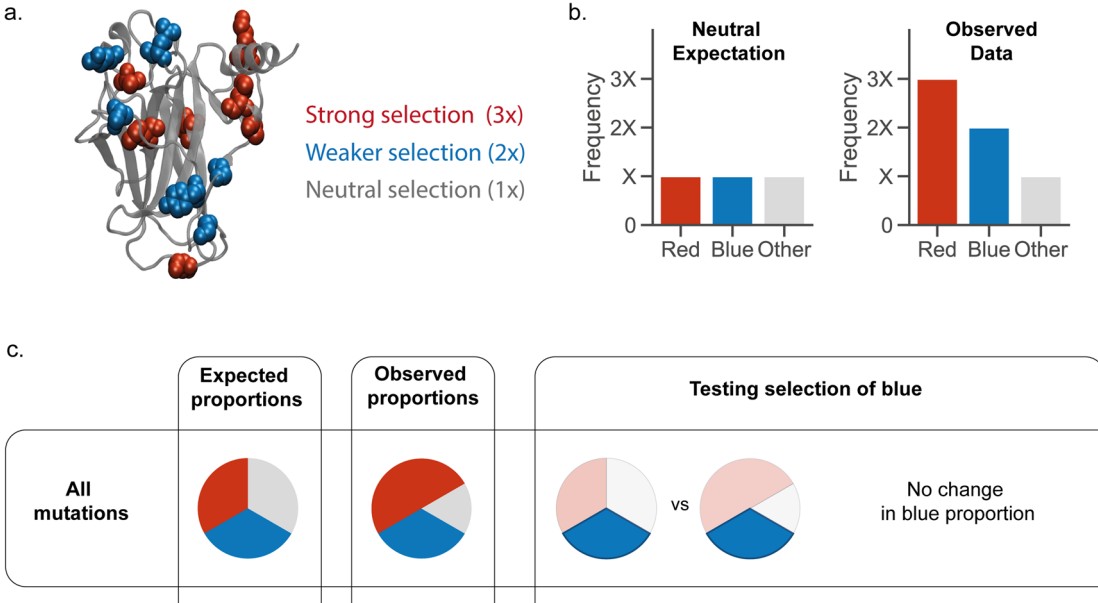

**Fig. 1 Detecting multiple selected mutation features. a** An example of protein with two types of mutations under selection. **b** The red mutations are under strong selection (they occur three times as often in the data than would be expected under neutral evolution), the blue mutations under weak selection (they occur twice as often as expected under neutral evolution), and the rest of the protein is neutral. **c** The overall proportion of strongly selected red mutants is increased compared to the expected proportions. However, the blue mutants appear in the same proportion as the neutral expectation. The increase in blue mutations is masked by the larger increase in red mutants. The red mutations can be excluded from both the expected model and the observed data to detect selection of the blue mutants.

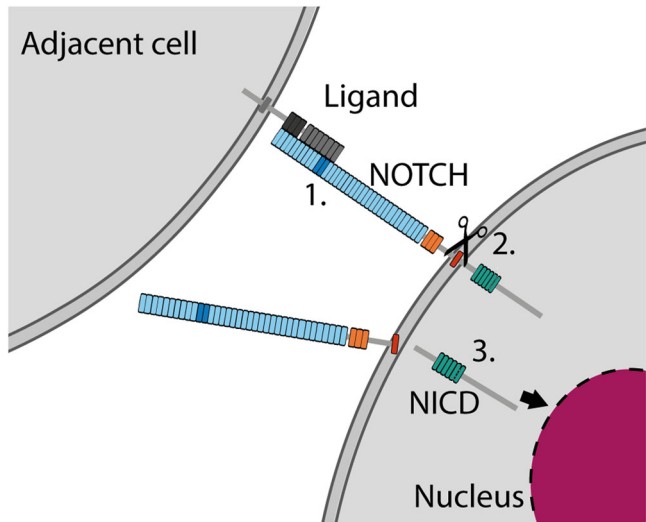

**Fig. 2 Mechanism of NOTCH activation.** 1 NOTCH is activated when a ligand from an adjacent cell binds with a subset of the NOTCH EGF repeats. EGF repeats shown in blue, key ligand-binding region EGF11–12 shown in dark blue; Lin-12/Notch Repeats (LNR) region, orange; transmembrane region, red; ankyrin repeats, green. 2 Ligand binding triggers a series of cleavage events, resulting in cleavage of the NOTCH transmembrane region (red) by γ-secretase. 3 This releases the intracellular domain of NOTCH (NICD) which travels to the nucleus to form part of a transcription factor.

In both tissues, studies of aged, non-cancerous epithelium detected high dN/dS ratios (an indication of positive selection) for both missense and nonsense mutations in *NOTCH1*, and *NOTCH1* mutant clones covered large proportions of the aged tissues[1,6]. Studies in mouse oesophagus have also found that loss of *Notch1* function conveys a strong competitive advantage to clones in normal tissue[11,12].

NOTCH proteins are membrane-bound cell surface receptors (Fig. 2) in a pathway that regulates cell fate[13]. These genes are critical regulators of differentiation in development and adult tissues and activating mutations or mutants that block protein function are found in different cancers[14,15]. The extracellular domains of NOTCH proteins contain up to 36 epidermal growth factor (EGF) repeats (Fig. 2). Many of the EGF repeats bind to calcium ions, which add rigidity to the structure, help fix the relative orientation of adjacent EGF repeats[16], and are required for ligand binding[17]. NOTCH ligands, from the Delta-like and Jagged families[18], are expressed by adjacent cells and bind to a subset of the EGF repeats[13] (Fig. 2), with EGF11 and 12 of *NOTCH1* particularly crucial for ligand binding[13,19]. This binding triggers a cascade of proteolytic events, resulting in the cleavage of the NOTCH transmembrane helix and the release of the NOTCH intracellular domain (NICD), which travels to the nucleus and forms part of a transcription factor complex that increases the expression of NOTCH target genes[13] (Fig. 2).

The impact on ligand binding of mutations in *NOTCH1* EGF repeats 11 and 12 (EGF11–12) has previously been described[1]. This region contains the highest concentration of missense mutations in the gene (Fig. 3a and Supplementary Fig. S3a), with 308 and 905 missense mutations detected in normal oesophagus and normal skin, respectively[1,6] (total missense mutations in *NOTCH1*: 831 in oesophagus, 2701 in skin). Recurrently mutated residues in this region (those mutated at least 4 times in the normal oesophagus data) include cysteines in disulphide bonds, buried glycines and hydrophobic packing residues[1]. These would all be expected to affect the stability of the protein[20,21] and could

prevent the structure from folding into the correct shape to bind with the ligand[22,23]. We, therefore, use the ligand-binding region of *NOTCH1* to introduce the method and confirm that it can detect (and assign statistical significance to) the previously observed patterns of selection in the normal oesophagus. We also demonstrate how excluding known forms of selection can help to detect weaker selection or less frequently selected features.

The stability of a protein is determined by the protein folding free energy, ΔG, which is the difference in Gibbs free energy between the folded and unfolded form of a protein. A mutation may alter ΔG (this change is called ΔΔG) and therefore stabilise or destabilise the protein. We used FoldX[24] to calculate the ΔΔG for each possible single nucleotide missense mutation[25] in NOTCH1 EGF11–12 (Fig. 3b, Methods). We constructed a null model of 'neutral' selection, which assumes that the distribution of mutations in the region will depend solely on the mutational spectrum (Supplementary Fig. S1 and Supplementary Note 2). By comparing the distribution of ΔΔG values from the observed mutations with the distribution expected under the null hypothesis (Supplementary Fig. S1), we found a significant enrichment of destabilising mutations with high ΔΔG values (expected median = 0.99 kcal/mol, observed median = 3.49 kcal/mol, $p < 2e^{-5}$, n = 308, two-tailed Monte Carlo test, Supplementary Note 10 and Fig. 3c). However, we can see that many observed mutations, including recurrent hotspots, do not appear to be destabilising (Fig. 3b and Supplementary Note 4). This suggests that some mutations are selected for reasons other than destabilising the protein structure.

Another mechanism expected to inactivate NOTCH1 ligand binding is disruption of the ligand-binding interface. We found that the observed proportion of missense mutations on the interface (Methods) was similar to the null model, and therefore the interface mutations were not significantly selected compared with the rest of EGF11–12 (Fig. 3d; expected = 35%, observed = 33%, p = 0.72, n = 308, two-tailed binomial test, Supplementary Note 10). However, this does not indicate that mutations on the interface are under neutral selection, just that they are not under stronger selection than the bulk of missense mutations in EGF11–12. Orthogonal mechanisms may dominate the selection landscape and make it more difficult to identify enrichment of interface mutations. As we have already shown that highly destabilising mutations are positively selected, all mutations with high ΔΔG values (with ΔΔG > 2 kcal/mol, Methods, results using a range of thresholds shown in Supplementary Fig. S2e) can be excluded from both the null model and the observed data and it can be tested whether, *within the non-destabilising mutations in EGF11–12*, there is an enrichment of interface mutations. Under this null model, there was a highly significant increase in the proportion of missense mutations on the ligand-binding interface (Fig. 3e, expected = 43%, observed = 64%, $p = 3e^{-5}$, n = 107, two-tailed binomial test, Supplementary Note 10).

After excluding destabilising mutations and those on ligand-binding sites, there are still some missense mutations remaining (Fig. 3f). Among these, the most frequent mutation in both skin and oesophagus is E455K, which forms part of a calcium-binding site that is known to be crucial for ligand binding[17,26]. Calcium binding is critical for the structural integrity of the EGF repeats[16,17], however, FoldX predicts that many mutations affecting the calcium-binding sites would not be destabilising (Fig. 3f). This may indicate that ΔΔG, as calculated by FoldX, is not fully capturing the disruption that mutations on calcium-binding residues are having on the protein structure. We, therefore, used MetalPDB[26] to define the calcium-binding residues in the structure of NOTCH1 EGF11–12 (Supplementary Fig. S2b, Methods) and tested for enrichment of mutations on these sites. Similar to the results for the interface mutations, testing with all mutations in

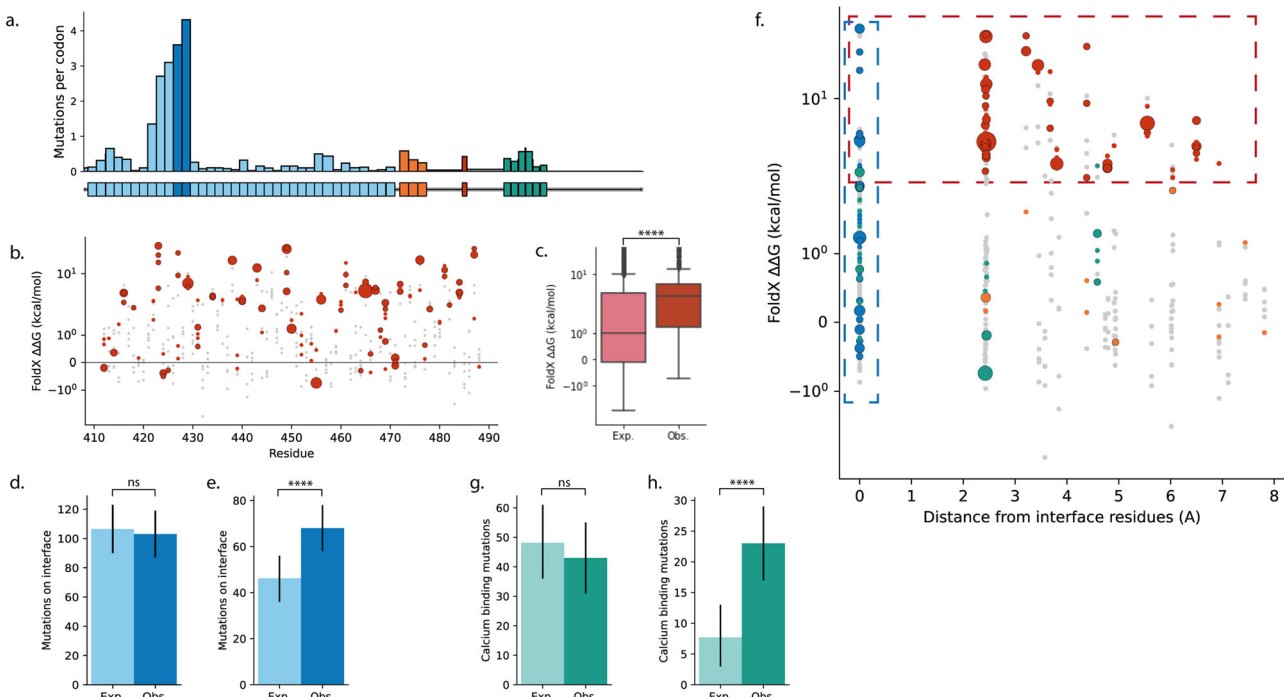

**Fig. 3 Patterns of selection of missense mutations in NOTCH1 EGF11–12 in normal human oesophagus. a** Missense mutation frequency across the domains of *NOTCH1*. Domain definitions from UniProt[56]. Where the gap between domains is only a single residue, mutations from this residue are included in the subsequent domain. EGF repeats, blue; EGF11–12, dark blue; LNR, orange; transmembrane region, red; ankyrin repeats, green; other regions, grey. **b** ΔΔG of mutations in NOTCH1 EGF11–12. Single nucleotide missense mutations that occur in the normal oesophagus, red, with marker size proportional to the number of times that mutation occurs. Single nucleotide missense mutations that do not occur in the dataset shown in grey. **c** Distributions of ΔΔG values of missense mutations. Distribution expected under the neutral null hypothesis, light red, and the distribution observed, dark red. **d, e** Counts of NOTCH1 EGF11–12 mutations occurring on the ligand-binding interface under the neutral null hypothesis, light blue, and observed, dark blue. Null and observed counts including all missense mutations (**d**) or excluding destabilising mutations (ΔΔG > 2 kcal/mol) from both the null model and observed data (**e**). **f** ΔΔG plotted against distance from the NOTCH1 EGF11–12 ligand-binding interface residues. Observed single nucleotide missense mutations shown in green (calcium binding), blue (ligand binding), red (ΔΔG > 2 kcal/mol) or orange (other). Marker size is proportional to the number of times that mutation occurs. Single nucleotide missense mutations that do not occur in the data set shown in grey. Regions containing highly destabilising mutations (ΔΔG > 2 kcal/mol) and mutations on the ligand-binding interface shown with dashed red and blue boxes, respectively. **g, h** Counts of NOTCH1 EGF11–12 mutations that are on calcium-binding residues under the neutral null hypothesis, light green, and observed, dark green. **g** Null and observed counts including all missense mutations (**g**) and excluding destabilising mutations (ΔΔG > 2 kcal/mol) and ligand-binding interface mutations from both the null model and observed data (**h**). *P* values calculated using a two-tailed Monte Carlo test for **c** and a two-tailed binomial test for **d, e, g, h** (Supplementary Note 10). Error bars in **d, e, g, h** show 95% confidence intervals (Supplementary Note 10). ****$P \leq 0.0001$, ns $P > 0.05$.

EGF11–12 did not detect a significant enrichment of mutations on the calcium-binding residues (expected = 16%, observed = 14%, $p = 0.48$, $n = 308$, two-tailed binomial test, Supplementary Note 10, Fig. 3g). However, by excluding the FoldX-destabilising (ΔΔG > 2 kcal/mol) and ligand-binding interface mutations, we found that the calcium-binding mutations are highly selected compared to the remaining mutations in the region (expected = 20%, observed = 59%, $p = 8e^{-8}$, $n = 39$, two-tailed binomial test, Supplementary Note 10 and Fig. 3h). Applying the analysis to mutations in normal skin reveals similar and highly significant selection of the same three categories of missense mutations: destabilising mutations, and mutations on the ligand-binding interface and on calcium-binding residues (Supplementary Fig. S3, this data previously analysed with this method in ref. [6]).

This example of missense mutations in *NOTCH1* EGF11–12 confirms that the statistical method can detect selection of known functional consequences of mutations, and can separate selection of different features in the same region (Supplementary Figs. S2d–f, S3e–g and Table S1). 95% and 89% of observed missense mutations in oesophagus (Fig. 3f and Supplementary Fig. S2c) and skin (Supplementary Fig. S3h, i) respectively are within the three categories of mutational impact examined here. The small proportion of mutations remaining may be weakly selected or

neutral passenger mutations, may be marginally outside of the category definitions used here, or may be selected due to a functional impact not tested here. For example, the most frequent mutation in *NOTCH1* EGF11–12 in skin that is not in the above categories is P460L (34 mutant clones), which has a ΔΔG value marginally below the chosen threshold of 2 kcal/mol (Supplementary Fig. S3i). The next most frequent uncategorised mutation, D464N (17 mutant clones) does not clearly belong to any of the three categories (Supplementary Fig. S3h, i), but may affect ligand-binding through its interaction with post-translational modifications[22,27]. The repeating method of testing for selection, excluding selected categories and testing again for new selected features may be useful for quickly classifying the majority of mutations and identifying outliers for further investigation.

**Using somatic mutations to generate protein-function hypotheses.** For *FBXW7*, the interpretation of the selected missense mutations is less straightforward. FBXW7 is one component of the Skp, Cullin, F-box containing E3 ubiquitin ligase complex[28]. FBXW7 recognises the substrate to be targeted for ubiquitination and subsequent degradation[28]. It has around 90 target substrates, including TP53 and NOTCH1[29].

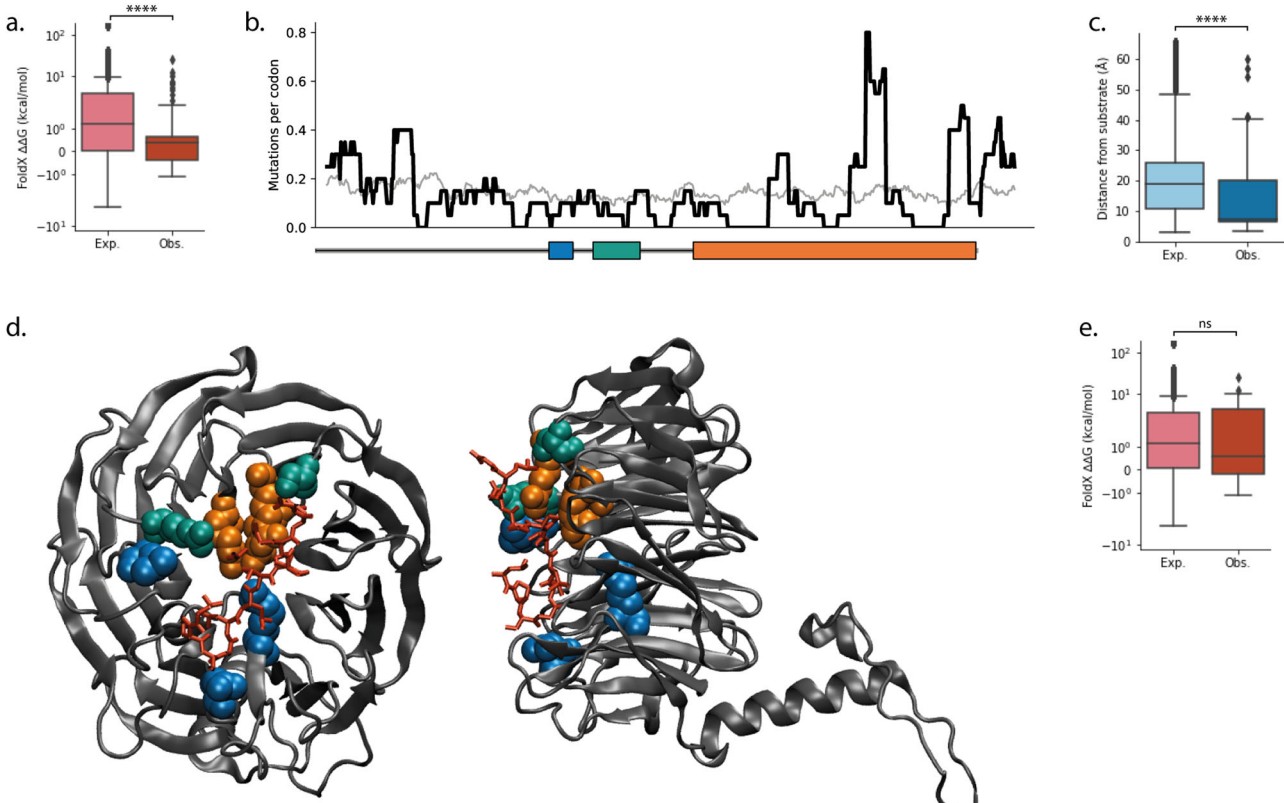

**Fig. 4 Missense mutations in *FBXW7*. a** Distribution of calculated ΔΔG values of missense mutations in FBXW7 in normal skin. Distribution expected under the neutral null hypothesis, light red, and the distribution observed, dark red. **b** Sliding window of missense mutation frequency across *FBXW7* in normal skin. Observed distribution shown with a bold black line, the expected distribution based on the mutational spectrum shown with a thin grey line. Dimerization domain, blue bar; F-box domain, green bar; WD40 domain, orange bar. **c** Distribution of distances of missense mutations in skin from the FBXW7 substrate (in this instance a 12-residue section of Cyclin E in PDB 2OVQ[30]). Distribution expected under the neutral null hypothesis, light blue, and the distribution observed, dark blue. **d** Structure of the FBXW7 WD40 domain (PDB 2OVQ[30]) bound to Cyclin E (red). Commonly mutated residues are highlighted. Blue and green residues contain at least four missense mutations in the normal skin dataset. Green residues also contain at least two missense mutations in normal oesophagus dataset. The three main hotspot residues in cancer (R465, R479 and R505) shown in orange. **e** Distribution of calculated ΔΔG values of missense mutations in skin, where all mutations within 8 Å of the FBXW7 substrate have been excluded from both the null model and the observed data. Distribution expected under the neutral null hypothesis, light red, and the distribution observed, dark red. *P* values in **a**, **c**, **e** calculated using the Monte Carlo test (Supplementary Note 10). ****$P \leq 0.0001$, ns $P > 0.05$.

There is some evidence of selection against loss-of-function mutations in normal skin, where nonsense and splice mutations are negatively selected (dN/dS = 0.31, $p = 0.025$, Supplementary Fig. S9[6]) and there is a significant deficit of destabilising missense mutations (expected median = 1.17 kcal/mol, observed median = 0.39 kcal/mol, $p = 0.0001$, $n = 62$, two-tailed Monte Carlo test, Supplementary Note 10 and Fig. 4a). However, the observed missense mutations are located significantly closer to the substrate binding site[30,31] than expected under the neutral null hypothesis (expected median = 18.7 Å, observed median = 7.3 Å, $p < 2e^{-5}$, $n = 62$, two-tailed Monte Carlo test, Supplementary Note 10 and Fig. 4b–d), suggesting that there is positive selection of a change to wild type *FBXW7* function. Re-testing the skin data while excluding mutations near the substrate binding site finds that there is no significant selection for destabilising mutations (excluding mutations within 8 Å of the FBXW7 substrate; expected median = 1.11 kcal/mol, observed median = 0.56 kcal/mol, $p = 0.55$, $n = 30$, two-tailed Monte Carlo test, Supplementary Note 10 and Fig. 4e). This suggests that the apparently strong negative selection of destabilising mutations seen when testing with the full set of missense mutations may, in fact, be partly a result of positive selection of non-destabilising mutations around the binding site. The peak of missense mutations in the oesophagus mirrors that of the skin data (Supplementary Fig. S10a

and Fig. 4b), although the small number of *FBXW7* mutations in the oesophagus data set means the statistical tests are non-significant (FoldX ΔΔG: expected median = 1.10 kcal/mol, observed median = 0.73 kcal/mol, $p = 0.59$, $n = 19$, Supplementary Fig. S10b; distance to substrate binding site: expected median = 18.7 Å, observed median = 13.2 Å, $p = 0.096$, $n = 19$, Supplementary Fig. S10c; two-tailed Monte Carlo test, Supplementary Note 10).

The most common *FBXW7* mutation hotspots in cancer – R465, R479 and R505[28] – are also located at the substrate binding site[30] (Fig. 4d). Mutations to these residues abrogate the ability of FBXW7 to bind to its substrates[28,31]. Inactivating *FBXW7* mutations are common in cancers driven by *NOTCH1* activating mutations, such as T-cell acute lymphoblastic leukaemia (T-ALL) and chronic lymphocytic leukaemia (CLL)[31]. The mutant FBXW7 cannot bind to NOTCH1 NICD, leading to an accumulation of NICD similar to that caused by *NOTCH1* activating mutations[31]. However, in the normal epithelia, there is strong selection for *NOTCH1* loss-of-function mutations, so selection of mutations which *increase* NOTCH1 activity would be surprising. In fact, despite their similar location in the protein structure, there is no overlap between the missense mutations that appear in the normal tissues and those that appear in T-ALL and CLL (COSMIC v91[32], T-ALL $n = 89$, CLL $n = 8$, normal skin

$n = 102$, normal oesophagus $n = 20$). This does not appear to be due to the mutational spectrum, as the most common cancer hotspot mutations (R465C, R465H, R479Q, R505C, combined total of zero occurrences in the normal skin) together would be expected to be mutated at least as often as the most frequently observed mutation in the normal skin, L559F (11 occurrences) (Supplementary Fig. S10d). Therefore, in the normal skin, there does not appear to be selection for loss of FBXW7 binding to NOTCH1 NICD.

It is tempting to speculate that, since *FBXW7* loss of function is a driver alongside *NOTCH1* gain-of-function mutations in T-ALL and CLL, selection of *NOTCH1* loss-of-function mutations in the normal epithelia would be accompanied by selection of gain-of-function *FBXW7* mutations. If *FBXW7* mutations lead to an increase in ubiquitination and degradation of *NOTCH1* NICD, then those mutant clones might have a growth advantage due to a reduction in *NOTCH1* signalling. However, due to the large number of *FBXW7* target substrates, several of which are driver genes in the normal epithelia, it is hard to narrow down to a single hypothesis. For example, an accumulation of c-MYC due to *Fbxw7* loss has been found to increase proliferation in keratinocytes in mice[33]. However, those cells also differentiate earlier due to accumulation of NOTCH1 NICD[33]. If the mutations in normal skin and oesophagus selectively abrogate FBXW7–c-MYC binding without disrupting FBXW7–NOTCH1 binding it may lead to clonal expansion.

**Distinct patterns of mutation in fitness-suppressing and fitness-enhancing proteins**. The two genes we examined above, *NOTCH1* and *FBXW7*, have very different relationships with cell fitness in the normal epithelia, and this is reflected in the patterns of selected mutations. Reducing NOTCH1 function increases cell fitness, as demonstrated by the strong positive selection of nonsense and essential splice (truncating) mutations (Supplementary Fig. S9a) and the widespread enrichment of missense mutations which disrupt NOTCH1 function through misfolding or preventing activation of NOTCH1 by its ligands. In contrast, an overall reduction of FBXW7 function reduces cell fitness, as shown by the negative selection of truncating mutations (Supplementary Fig. S9a). The missense mutations in this protein are highly concentrated in a key functional site, with very few mutations that are likely to cause major disruptions to the protein structure.

We therefore propose that, based on their relationship with cell fitness in the normal tissue, genes can be categorised in two groups: fitness-suppressors and fitness-enhancers. In fitness-suppressors, loss-of-function mutations lead to an increase in cell fitness, whereas in fitness-enhancers, activating mutations lead to greater cell fitness. The pattern of mutational selection will depend on which of these two groups the gene belongs to (Fig. 5a, b). Mutations to functional sites may increase, alter or decrease the protein function, and, therefore, may be enriched in both fitness-suppressors and fitness-enhancers (Fig. 5a, b). In contrast, mutations which cause general disruption to the protein structure, such as destabilising missense mutations in the protein core (Supplementary Fig. S14), frequently cause loss of protein function[34] and are therefore enriched in the fitness-suppressors (Fig. 5a, b).

To test this hypothesis further, we examined the selection of missense mutations in a few more well characterised proteins at both ends of the spectrum of truncating mutation selection (Supplementary Fig. S9a). Like *NOTCH1*, the genes *NOTCH2* and *TP53* are under strong positive selection for loss-of-function truncating mutations (Supplementary Fig. S9a and Supplementary Notes 6 and 7), and are therefore fitness-suppressors. NOTCH2 is a highly similar protein to NOTCH1, with the same mechanism of activation (Fig. 2). Using our method to analyse missense mutations in the key ligand-binding EGF repeats of *NOTCH2* reveals the same statistically significant patterns of selection as found in EGF11–12 of *NOTCH1*: positive selection of the structure-disrupting calcium-binding- and destabilising mutations, along with positive selection of mutations on the ligand-binding interface (Fig. 5b, Supplementary Note Section 5 and Supplementary Fig. S11). Similarly, in the DNA-binding domain of the *TP53* transcription factor, there is highly significant selection of destabilising mutations and mutations close to the interface with the DNA molecule (Fig. 5b, Supplementary text Section 6 and Supplementary Fig. S12). The patterns of missense mutations in these two proteins are therefore consistent with those expected for fitness-suppressor proteins (Fig. 5a, b).

In contrast, truncating mutations in the kinase encoding gene *PIK3CA* are under strong negative selection (Supplementary Fig. S9a and Supplementary Note 8). In the normal skin, dN/dS analysis finds no significant selection of *PIK3CA* missense mutations (Supplementary Fig. S9b). However, applying our statistical method to *PIK3CA* reveals a highly significant enrichment of previously identified activating missense mutations (Supplementary Note 7 and Supplementary Fig. S13b, c), and positive selection of mutations in key regions of the protein known to contain activating mutations, including on the interface with the inhibitory p85α protein, on the link between the adaptor-binding domain and the Ras-binding domain, and in the kinase domain (Supplementary Note 7, Supplementary Fig. S13d–i and Tables S3, S4)[6]. This concentration of missense mutations in a few functional sites is consistent with the pattern expected for a fitness-enhancer protein.

To further confirm this relationship between patterns of missense mutation selection and the function the gene plays in cell fitness, we compared selection for truncating mutations with selection of protein destabilising mutations. Among the genes sequenced in the normal skin, there is a strong correlation between selection of destabilising missense mutations and selection of truncating mutations (Fig. 5c, Pearson's correlation coefficient $= 0.89$, two-tailed $p = 8e^{-6}$, $n = 15$). This pattern is also seen in mutations in mutagen-treated mouse oesophagus (Supplementary Fig. S15, Supplementary methods, Pearson's correlation coefficient $= 0.64$, two-tailed $p = 6e^{-11}$, $n = 84$). This would suggest that an analysis of the impact of mutations on folding can support classification into either fitness-suppressors or fitness-enhancers.

## Discussion

Here, we have adapted a statistical method for cancer driver gene discovery to look for selected features of mutations in a gene. By using this method, protein structural and functional information can be drawn from the increasingly large amount of DNA sequencing data available. We have found biologically plausible and statistically significant patterns of selection in several proteins. This approach can associate mutational changes in protein structure or function with cell fitness, even in the absence of 'hotspot' mutations and in the presence of passenger mutations. For example, in *NOTCH2* EGF11−12, no mutations occurred more than twice in the normal oesophagus dataset. However, by considering those mutations in bulk, we have identified three statistically significant features of selected missense mutations (the same features that are selected for in *NOTCH1* EGF11−12).

Manual investigation of hotspot mutations can provide similar information to our analysis[1] but has disadvantages: it can be time-consuming, does not leverage the information provided by rarer mutations and does not statistically test the selection of mutation features. In vitro mutational scanning experiments that

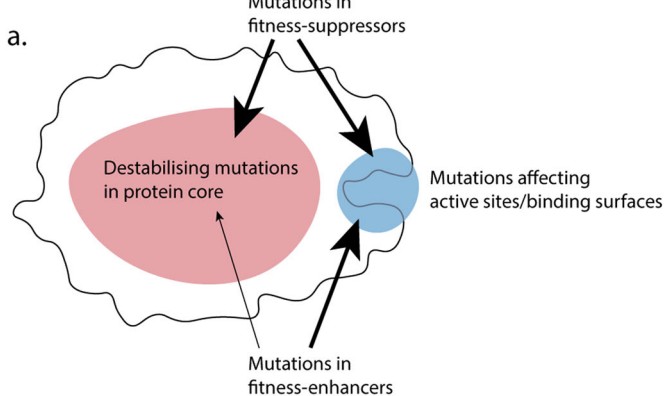

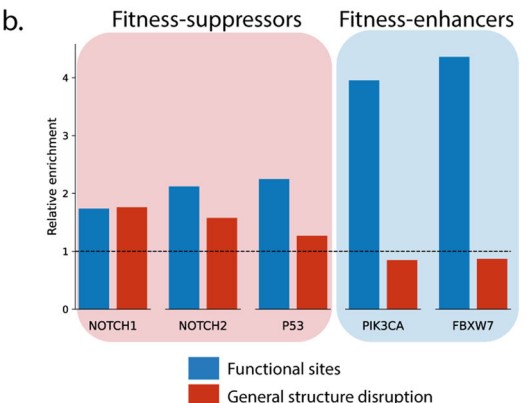
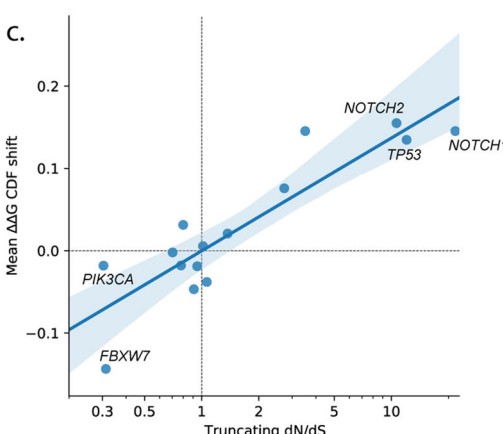

**Fig. 5 General patterns of mutations in fitness-suppressing and fitness-enhancing proteins. a** Common loss-of-function missense mutations disrupt critical functional sites or cause larger scale destabilisation of the protein structure, while common gain-of-function mutations affect critical functional sites. **b** Both fitness-suppressor and fitness-enhancer proteins in normal skin show an enrichment of mutations on functional sites compared to the null model (excluding structure-disrupting mutations from both the null model and observed data). In fitness-suppressors only, there is an enrichment of mutations that generally disrupt the protein structure (excluding mutations on functional sites). Functional sites are defined as the ligand-binding interface for NOTCH1 and NOTCH2, residues within 8 Å of DNA or substrate for TP53 and FBXW7 respectively, and residues in the regions 100–119, 444–473, 962–980 for *PIK3CA*. Structure-disrupting mutations are defined as mutations with $\Delta\Delta G > 2$ kcal/mol and mutations on calcium-binding residues in NOTCH1 and NOTCH2. **c** Selection of destabilising missense mutations is strongly correlated with selection of nonsense mutations in normal skin. Selection of destabilising missense mutations is shown as the shift in the distribution of $\Delta\Delta G$ values between the null model and the observed data. CDF values were used instead of raw $\Delta\Delta G$ values to reduce the influence of extreme outliers (Supplementary Note 10). $\Delta\Delta G$ CDF shift is the difference between the mean of the null and observed distributions of $\Delta\Delta G$ CDF values. It is a value between $-0.5$ (all observed mutations have the minimum $\Delta\Delta G$ possible in the protein structure) and 0.5 (all observed mutations have the maximum $\Delta\Delta G$ possible). A $\Delta\Delta G$ CDF shift of 0 means the mean values of the null and the observed $\Delta\Delta G$ CDF distributions are equal. Structures of wild type proteins containing at least 50 missense mutations in the skin dataset were used (Supplementary Note 10), and the average $\Delta\Delta G$ CDF shift of all structures for each gene was calculated. Pearson's correlation coefficient $= 0.89$, two-tailed $p = 8e^{-6}$, mean $\Delta\Delta G$ CDF shift vs logarithm of truncating dN/dS ratio. Blue line and shaded region show the linear regression and 95% confidence interval of the regression estimate calculated and plotted using the Python package Seaborn[57].

exhaustively mutate a protein or protein domain and measure the phenotypic consequences of each mutation[35] can provide information both about the effects of individual mutations and about protein function and structure[35]. However, these experiments are typically carried out in culture[35], an environment which can substantially alter cell phenotype[36,37]. *In-silico* mutational scanning has become increasingly common as computational power increases and *in-silico* methods improve[25,38–40]. This technique predicts the effects of mutations on protein function, but relating molecular change to alterations in cell function is not trivial[41]. Our analysis method effectively combines *in-silico* mutational scanning with in-vivo mutational phenotype assays that are the DNA-sequencing datasets of normal tissues.

Some caution must be applied when interpreting the results of the method. Although protein misfolding and ligand binding are well-known processes that control protein activity[24,42],

correlation does not mean causation, and significant selection may be found for a feature that correlates (coincidentally or otherwise) with the true selected feature. This is also a test for selection *relative to the rest of the tested region*, and does not necessarily directly translate to positive or negative selection. We have shown that if there are multiple selected features in a region, testing for one individual feature at a time may lead to misleading results, but that this can be corrected by accounting for the other, confounding selected features.

For proteins that are sufficiently ordered that their structures can be determined, selection of loss-of-function nonsense and splice mutations is frequently accompanied by selection of destabilising missense mutations. A standard workflow for analysing missense mutations in these genes might start by identifying if destabilising mutations are selected for before searching for other selected mutation features. Many different mutations in

Human    DVDEC**S**LGANPCEHAGKC**I**NTLGSFECQCLQGYTGPRCEIDVNEC**V**SNPCQNDATCLDQIGEFQCICMPGYEGV**H**CE

Rat      DVDEC**A**LGANPCEHAGKC**L**NTLGSFECQCLQGYTGPRCEIDVNEC**I**SNPCQNDATCLDQIGEFQCICMPGYEGV**Y**CE

**Fig. 6 Alignment of EGF11–12 of human and rat NOTCH1 protein sequences.** Residues 412 to 488 of each sequence shown, residues which differ highlighted in yellow.

| Table 1 Ligand-binding interface residues of NOTCH1 EGF11−12 and NOTCH2 EGF11−12. | |
| --- | --- |
| | **Ligand-binding interface residues** |
| NOTCH1 EGF11−12 | 413, 415, 418, 420, 421, 422, 423, 424, 425, 435, 436, 444, 447, 448, 450, 451, 452, 454, 466, 467, 468, 469, 470, 471, 475, 477, 478, 479, 480 |
| NOTCH2 EGF11−12 | 418, 421, 424, 425, 426, 428, 429, 439, 440, 452, 454, 456, 470, 472, 473, 481 |

NOTCH1 ligand-binding interface residues based on Supplementary Fig. S3 of ref. [23].
Ligand-binding interface residues of NOTCH2 EGF11−12 from ref. [48].

| Table 2 Calcium-binding residues of NOTCH1 EGF11−12 and NOTCH2 EGF11−12. | |
| --- | --- |
| | **Calcium-binding residues** |
| NOTCH1 EGF11−12 | 412, 413, 415, 431, 432, 435, 452, 453, 455, 469, 470 |
| NOTCH2 EGF11−12 | 415, 416, 418, 435, 436, 439, 456, 457, 459, 473, 474 |

Based on MetalPDB[26] and the structure 2VJ3[49] for NOTCH1 and structure 5MWB[48] for NOTCH2.

a gene may disrupt protein function, but only a small proportion of potential mutations are likely to be gain-of-function[43]. This means that selection for gain-of-function mutations in a gene can lead to hotspots or small clusters of mutations[43], such as those seen in *PIK3CA* and *FBXW7*. However, as demonstrated by the pattern of mutations in *NOTCH1*, clusters of mutations and hotspot mutations are also seen in genes where loss-of-function mutations are selected. Selection for gain-of-function mutations may be accompanied by selection against loss-of-function mutations, such as nonsense mutations or missense mutations that cause misfolding. This may dilute the signs of selection when analysing missense mutations across the gene as a whole, meaning the selection on these genes may be harder to detect using some driver detection methods.

Altogether, the analysis presented here demonstrates a route to mine the rich vein of information available in large DNA sequencing data sets through the integration of features from other domains, such as protein structure. Through combining diverse datasets, we can infer the selection of functional changes in proteins, and hence learn about both the protein structure −function relationship and the role of the protein in the tissue sequenced. The method can be used as an in-vivo validation of results of in-vitro studies, and could be a useful method to explore selection of mutation features in existing data sets prior to conducting further experiments. This is an approach that will be widely applicable for genes or protein domains that are positively or negatively selected in somatic contexts, whether in cancer or normal tissue.

## Methods

**Data**. We analysed mutations detected in normal human oesophagus[1], normal human skin[6], and mutagen-treated mouse oesophagus[11]. All studies used DNA sequencing to detect mutations in a grid of adjacent tissue samples. Large clones could spread over multiple samples. To avoid double counting of such clones, we used the mutations list where mutations that were seen repeatedly in nearby samples were assumed to be from a single clone and were merged[1,6,11].

**FoldX**. We used FoldX 5 to calculate the ΔΔG values of mutations. For each PDB file, the FoldX command *RepairPDB* was run to minimise steric clashes and optimize residue orientation. Then the FoldX command *PositionScan* was run for

every residue of the protein chains of interest in the structure. This command mutates each residue to all other amino acids and calculates the ΔΔG value for each mutation. Default FoldX settings were used for both the *RepairPDB* and *PositionScan* commands.

Some analyses required a threshold to discriminate destabilising mutations from non-destabilising mutations. Unless otherwise noted, we used a threshold ΔΔG value of 2 kcal/mol because this has been used in previous studies to define mutations which are highly destabilising[44–47].

**Ligand-binding interface residues**. The ligand-binding interface residues in EGF11 and EGF12 have been identified for the rat NOTCH1 bound to the ligands JAG1 and DLL4[22,23]. *NOTCH* genes and ligands are highly conserved between species meaning that the results from the rat protein can be applied to human NOTCH1[48] (Fig. 6). The ligand-binding surface is very similar for both ligands[23] and we, therefore, chose to use the union of both sets of ligand-binding residues (Supplementary Fig. S2a and Table 1). The ligand-binding residues for NOTCH2 are based on conservation with the rat NOTCH1 ligand interface[48] (Table 1).

**Calcium-binding mutations**. The calcium-binding residues in EGF11−12 of NOTCH1 were defined using MetalPDB[26] and the 2VJ3 structure[49] (Table 2). The calcium-binding residues in EGF11−12 of NOTCH2 were defined using MetalPDB[26] and the 5MWB structure[48] (Table 2).

**Conservation scores**. Phylop conservation scores[50] for each position were downloaded via the UCSC genome browser, http://hgdownload.soe.ucsc.edu/goldenPath/hg19/phyloP100way/[51].

**Images of protein structures**. VMD[52] was used to visualise protein structures and images were rendered using Tachyon[53].

**Principle of the statistical method**. Cells acquire mutations as a result of exposure to mutagens (e.g., tobacco, alcohol or ultraviolet light) or cell-intrinsic processes[54]. The pattern of mutations in a sample, known as the mutational spectrum, will depend on the particular mutational and DNA repair processes involved[54]. The chance of acquiring a specific single nucleotide substitution appears to depend (at least partially) on the adjacent nucleotides[54].

The probability of a particular somatic mutation appearing in a sequenced sample depends on both the rate at which the mutation occurs in cells and the strength of selection on the mutation once it has occurred (Supplementary Note 3). In genes where no mutations of any kind convey a growth advantage/disadvantage, the mutations detected by DNA sequencing will simply be an unbiased sample of the mutations produced by the spectrum. Mutations which convey a growth advantage to the cell are more likely to appear in multi-cellular clones that are detected in the sequencing data, and mutations that convey a disadvantage are less likely to be observed.

The model of neutral null hypothesis is created by combining the mutational spectrum for the dataset (Methods 5.8) and assigning an expected probability of occurrence to each mutation. Those probabilities are then assigned to the mutation scores (which could be ΔΔG, distance from a substrate site, whether the mutation sits on an ion-binding site etc.) to construct an expected distribution of those scores (Supplementary Fig. S1). The observed distribution of scores can then be statistically compared (Supplementary Note 10) to the expected distribution (Supplementary Fig. S1).

**Mutational spectrum calculation.** Firstly, all genes containing an exonic mutation in the data sets were found using exon locations in GRCh37.p13 downloaded from Ensembl Biomart[55]. For each of these genes, the longest transcript was selected and alternative transcripts discarded. A trinucleotide context was calculated in the direction of the protein transcription for every nucleotide in the coding sequence of each transcript and applied to each observed mutation. For each data set, a mutation rate was calculated for each single nucleotide substitution type in each trinucleotide context by dividing the total number of observed mutations of that trinucleotide change by the number of times the trinucleotide context occurs in the included transcripts. Further information is given in Supplementary Note 2.

**Statistics and reproducibility.** Throughout this article, Binomial and Monte Carlo tests have been used to calculate $p$ values, depending on whether the score being tested is Boolean or not. Analyses on NOTCH1 were performed on either 831 and 2701 missense mutations (normal oesophagus and normal skin, respectively), or subsets of those mutations as described in the text. A full description about the tests used and their interpretation is given in Supplementary Note 10, and notes on rerunning analyses are linked in Supplementary Note 9.

**Reporting summary.** Further information on research design is available in the Nature Portfolio Reporting Summary linked to this article.

## Data availability

The merged mutation data from normal oesophagus is available in Supplementary Table 2 of ref.[1] in the sheet named 'Mutations_collapsed_by_distance'. The merged mutation data from normal skin is available in Supplementary Table S4 of ref.[6]. For both datasets, we used all exonic single-nucleotide mutations in those tables. In addition, a full list of dN/dS results for the skin data was created by running the R package dndscv[9] (https://github.com/im3sanger/dndscv) on the data in Supplementary Table S4. The mouse oesophagus mutation data are available in Supplementary Table 2 of ref.[11]. We used the mutations from the three mice treated with the mutagen diethylnitrosamine. Source data for figures can be found in Supplementary Data 1.

## Code availability

The software used to run the statistical analysis and produce plots is available here: https://doi.org/10.5281/zenodo.8077427 Jupyter Notebooks for generating the figures are available at https://doi.org/10.5281/zenodo.8077429 Instructions on rerunning the calculations are given in Supplementary Note 9.

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

## Acknowledgements
This work was supported by grants from the Wellcome Trust to the Wellcome Sanger Institute (098051 and 296194) and Cancer Research UK Programme Grants to P.H.J. (C609/A17257 and C609/A27326). B.A.H. and M.W.J.H. are supported by the Medical Research Council (Grant-in-Aid to the MRC Cancer unit grant number MC_UU_12022/9 and NIRG to B.A.H. grant number MR/S000216/1). M.W.J.H. acknowledges support from the Harrison Watson Fund at Clare College, Cambridge. B.A.H. acknowledges support from the Royal Society (grant no. UF130039).

## Author contributions
M.W.J.H, D.S. and B.A.H. conceived the study. M.W.J.H. wrote the software and performed statistical analysis. R.A. calculated misfolding energies for mouse proteins. M.W.J.H. wrote the manuscript with input from all authors. B.A.H. and P.H.J. supervised the research.

## Competing interests
The authors declare no competing interests.
