## [Peer Review File · Communications Biology]

Reviewers' comments:

Reviewer #1 (Remarks to the Author):

The paper presents a statistical-based method to uncover specific structural features in somatic mutations. Such a method is able to explore regions with weak subsets of mutations that are important in a biological perspective but could be hidden for widespread mutations. Such a feature differs the proposed approach from other tools.

The results show selection of specific structural features in drivers and how these drivers may be classified as fitness-enhancing and fitness-suppressing through different patterns of mutation enrichment.

The paper is well written and easy to follow. The authors present experiments and robust results from the application of the proposed method. Such experiments validate the application of the method within the application context. Important aspects of experiments are presented, especially in the supplementary material.

I emphasize that the authors were concerned with making available the codes of the experiments that generated the results, analyses and plots. This is of fundamental importance to allow the reproducibility of the experiments and allow future and related works to compare the results with the proposed approach.

Suggestions:

- The results are interesting, but how can they be compared with related methods? A “Related Work” section could be presented and similar methods could be compared with the proposed approach. In Section 1 of Supplementary Text the authors state that the proposed method differs from the other methods in two key ways. However, it could be interesting to cite related methods and perform a qualitative and quantitative comparison
- I understand that input data is available in the original publications. However, such input data could be described in Supplementary Material as described in the code repository.
- For the easy understanding, would it be possible to insert in the supplementary material a small section where the method is run in a small example? I understand the difficulty in meeting this point, but it is something that would help readers who are not familiar with statistical methods and are interested in the approach

Reviewer #2 (Remarks to the Author):

In this study, the authors present a flexible statistical method to investigate the selection of structural features in datasets of somatic mutants in normal tissues. They demonstrate how this approach can identify specific structural features under selection in key driver genes within aged tissues. Furthermore, the authors demonstrate the classification of drivers as either fitness-enhancing or fitness-suppressing based on different patterns of mutation enrichment.

Overall, this study offers a valuable pathway for understanding the mechanisms of protein function through in vivo mutant selection. The authors propose that somatic missense mutations can provide valuable insights into the relationship between protein structure, function, and cell fitness. However, it

would be beneficial if the authors provide more detailed explanations about the specific datasets used and the rationale behind the selection of NOTCH1 and FBXW7 as the focus of the study.

Comments:

(1) I acknowledge that the data sources are derived from previous studies (1, 6). Nevertheless, in the context of this study, it is imperative for the authors to provide sufficient explanations regarding the data, including details on the specific samples utilized or whether all samples were encompassed. Additionally, while the abstract mentions aging tissues, it is necessary for the authors to provide a corresponding description of the data within the main text.

(2) The author's statement " NOTCH1 is a strong driver of clonal expansion in normal skin and oesophageal epithelium (1, 6)" is supported by the studies referenced in literature 1 and 6. Nonetheless, in the context of this specific study, it would be advantageous for the authors to furnish more comprehensive details and elucidations pertaining to this conclusion.

(3) Please provide an explanation regarding the recurrently mutated residues mentioned in the sentence, "Recurrently mutated residues in this region include cysteines in disulphide bonds, buried glycines and hydrophobic packing residues (1)."

(4) The author extensively elaborated on "the null model" in the supplementary material, but provided limited explanation in the main text. It is suggested to incorporate necessary explanations regarding the null model in the methods section of the main text.

(5) Hotspot mutations in cancer were discussed specifically for FBXW7, but not for NOTCH1. What is the reason for this?

(6) "The peak of missense mutations in the oesophagus mirrors that of the skin data (Figure S10a, Figure 4d),..." There is a mislabeling here. It should be Figure 4b instead.

(7) Please check Figure 4B. According to the description, there should be two figures, and it seems that the author missed one.

Reviewer #1 (Remarks to the Author):

The paper presents a statistical-based method to uncover specific structural features in somatic mutations. Such a method is able to explore regions with weak subsets of mutations that are important in a biological perspective but could be hidden for widespread mutations. Such a feature differs the proposed approach from other tools.

The results show selection of specific structural features in drivers and how these drivers may be classified as fitness-enhancing and fitness-suppressing through different patterns of mutation enrichment.

The paper is well written and easy to follow. The authors present experiments and robust results from the application of the proposed method. Such experiments validate the application of the method within the application context. Important aspects of experiments are presented, especially in the supplementary material.

I emphasize that the authors were concerned with making available the codes of the experiments that generated the results, analyses and plots. This is of fundamental importance to allow the reproducibility of the experiments and allow future and related works to compare the results with the proposed approach.

Suggestions:

- The results are interesting, but how can they be compared with related methods? A "Related Work" section could be presented and similar methods could be compared with the proposed approach. In Section 1 of Supplementary Text the authors state that the proposed method differs from the other methods in two key ways. However, it could be interesting to cite related methods and perform a qualitative and quantitative comparison

Our proposed method serves a different purpose from previous driver detection methods, and therefore a quantitative comparison is not applicable. We have, however, expanded Section 1 of the supplementary text to explain in more detail the qualitative differences between our method and previous driver detection methods. The updated section is as follows (new/edited sentences highlighted):

"There are a number of methods that use a similar statistical principle to detect genes under selection. They assume that mutations are generated according to the mutational spectrum (Figure S1, methods 5.7-5.8). In genes where no mutations of any kind convey a growth advantage/disadvantage, the mutations detected by DNA sequencing will simply be an unbiased sample of the mutations produced by the spectrum. In genes under selection, it is assumed that certain types of mutations (missense/nonsense mutations (2), or mutations estimated by machine-learning approaches to have high functional impact (3, 4)) are more likely to alter cell phenotype than others (silent mutations, mutations with low estimated functional impact). Therefore, more high-impact mutations will be detected in a positively selected gene than would be expected under the neutral null hypothesis (2-4) (Figure S1). Conversely, negative selection in a gene will lead to the detection of fewer high-impact mutations than would be expected under the null hypothesis.

Our analysis method is based on the same principles as the cancer driver detection tools (Figure S1). However, it differs from the driver detection tools in two key ways. First, the method is set up to be flexible - allowing the selection of many different aspects of mutations to be tested within the same framework. Secondly, and crucially, the method is able to distinguish and separately test selection

due to different functional changes within the same region – as multiple forms of selection can obscure each other if not properly accounted for (Figure 1).

These differences mean that our tool performs a different function from the driver detection methods. These driver detection methods are designed to return a comprehensive-as-possible set of genes under selection, and to do so use scores that represent the overall severity of the functional impact of mutations. Our tool, in contrast, does not come with a default scoring method – you can test for any functional impact that can be represented as a numerical score for each mutation. Testing for a single functional impact will not detect drivers that are under selection for a different mutational effect, and testing the same dataset with multiple scores risks producing false positives.

Our method is best suited to detailed investigation of individual genes. Unlike the driver detection tools, our method can link selection with particular alterations to gene mechanisms and can separate multiple types of selected mutation in the same region. The tool can be used in combination with driver detection tools, which can narrow down candidate genes for detailed investigation, and prior knowledge can be incorporated to tailor the statistical analysis to the gene of interest.”

- I understand that input data is available in the original publications. However, such input data could be described in Supplementary Material as described in the code repository.

We thank the reviewer for this suggestion. We have added further descriptions of how to access the data, as was described in the code repository.

The methods section on data now reads as follows:

“We analysed mutations detected in normal human oesophagus (1), normal human skin (6), and mutagen-treated mouse oesophagus (11). All studies used DNA sequencing to detect mutations in a grid of adjacent tissue samples. Large clones could spread over multiple samples. To avoid double counting of such clones, we used the mutations list where mutations that were seen repeatedly in nearby samples were assumed to be from a single clone and were merged (1, 6, 11).

The merged mutation data from normal oesophagus is available in Supplementary Table 2 of (1) in the sheet named “Mutations_collapsed_by_distance”. The merged mutation data from normal skin is available in Supplementary Table S4 of (6). For both datasets, we used all exonic single-nucleotide mutations in those tables. In addition, a full list of dN/dS results for the skin data was created by running the R package dndscv (9) (<https://github.com/im3sanger/dndscv>) on the data in Supplementary Table S4. The mouse oesophagus mutation data is available in Supplementary Table 2 of (11). We used the mutations from the three mice treated with the mutagen diethylnitrosamine.”

- For the easy understanding, would it be possible to insert in the supplementary material a small section where the method is run in a small example? I understand the difficulty in meeting this point, but it is something that would help readers who are not familiar with statistical methods and are interested in the approach

We thank the reviewer for this question and agree that an example or two would be very helpful for readers who might be interested in trying the approach. We have created a set of guides to using the method here: https://github.com/michaelhall28/darwinian_shift/wiki

We have added the following section in the supplementary text to point the reader to this guide.

“Guide to running analyses

A more comprehensive guide to the method is available here:
https://github.com/michaelhall28/darwinian_shift/wiki.

This guide explains how to run basic and more complex analysis and includes numerous Python code examples.”

Reviewer #2 (Remarks to the Author):

In this study, the authors present a flexible statistical method to investigate the selection of structural features in datasets of somatic mutants in normal tissues. They demonstrate how this approach can identify specific structural features under selection in key driver genes within aged tissues. Furthermore, the authors demonstrate the classification of drivers as either fitness-enhancing or fitness-suppressing based on different patterns of mutation enrichment. Overall, this study offers a valuable pathway for understanding the mechanisms of protein function through in vivo mutant selection. The authors propose that somatic missense mutations can provide valuable insights into the relationship between protein structure, function, and cell fitness. However, it would be beneficial if the authors provide more detailed explanations about the specific datasets used and the rationale behind the selection of NOTCH1 and FBXW7 as the focus of the study.

We have expanded the introduction to describe the rationale behind the selection of NOTCH1 and FBXW7 (new text highlighted).

“Due to the large available sample size from these studies and the known functional impact of the frequently occurring mutations, we use NOTCH1 to introduce and validate the method. We also examine mutations in FBXW7 to explore mutational patterns and generate hypotheses linking protein function to cell fitness. **Between them, these two genes show the versatility and wide applicability of the method.**”

Comments:

(1) I acknowledge that the data sources are derived from previous studies (1, 6). Nevertheless, in the context of this study, it is imperative for the authors to provide sufficient explanations regarding the data, including details on the specific samples utilized or whether all samples were encompassed. Additionally, while the abstract mentions aging tissues, it is necessary for the authors to provide a corresponding description of the data within the main text.

We have expanded the description of the data in the methods to clarify that we used all samples from the two human studies, and the samples from the three mutagen-treated mice from the mouse oesophagus studies.

We have also added a sentence in the introduction to describe the data collected in the previous studies.

Intro text:

“We demonstrate the power of this method using datasets of mutant clones found through DNA sequencing of normal human oesophageal epithelium (1) and skin (6). **These studies collected samples of normal, non-cancerous tissues samples from individuals of different ages.**”

Methods text:

“We analysed mutations detected in normal human oesophagus (1), normal human skin (6), and mutagen-treated mouse oesophagus (11). All studies used DNA sequencing to detect mutations in a grid of adjacent tissue samples. Large clones could spread over multiple samples. To avoid double counting of such clones, we used the mutations list where mutations that were seen repeatedly in nearby samples were assumed to be from a single clone and were merged (1, 6, 11).

The merged mutation data from normal oesophagus is available in Supplementary Table 2 of (1) in the sheet named “Mutations_collapsed_by_distance”. The merged mutation data from normal skin is available in Supplementary Table S4 of (6). For both datasets, we used all exonic single-nucleotide mutations in those tables. In addition, a full list of dN/dS results for the skin data was created by running the R package dndscv (9) (<https://github.com/im3sanger/dndscv>) on the data in Supplementary Table S4. The mouse oesophagus mutation data is available in Supplementary Table 2 of (11). We used the mutations from the three mice treated with the mutagen diethylnitrosamine.”

(2) The author's statement " NOTCH1 is a strong driver of clonal expansion in normal skin and oesophageal epithelium (1, 6)" is supported by the studies referenced in literature 1 and 6. Nonetheless, in the context of this specific study, it would be advantageous for the authors to furnish more comprehensive details and elucidations pertaining to this conclusion.

We have expanded the description in the text.

“NOTCH1 is a strong driver of clonal expansion in normal skin and oesophageal epithelium (1, 6). In both tissues, studies of aged, non-cancerous epithelium detected high dN/dS ratios (an indication of positive selection) for both missense and nonsense mutations in NOTCH1, and NOTCH1 mutant clones covered large proportions of the aged tissues (1, 6). Studies in mouse oesophagus have also found that loss of NOTCH1 function conveys a strong competitive advantage to clones in normal tissue (11, 12).”

(3) Please provide an explanation regarding the recurrently mutated residues mentioned in the sentence, “Recurrently mutated residues in this region include cysteines in disulphide bonds, buried glycines and hydrophobic packing residues (1).”

We have clarified the definition of recurrently mutated residues:

“Recurrently mutated residues in this region (those mutated at least 4 times in the normal oesophagus data) include cysteines in disulphide bonds, buried glycines and hydrophobic packing residues (1).”

(4) The author extensively elaborated on "the null model" in the supplementary material, but provided limited explanation in the main text. It is suggested to incorporate necessary explanations regarding the null model in the methods section of the main text.

We thank the reviewer for this suggestion. We have moved the description of the statistical principles and the null model to the methods in the main text.

The new section in methods is as follows:

“Principle of the statistical method

Cells acquire mutations as a result of exposure to mutagens (e.g. tobacco, alcohol or ultraviolet light) or cell-intrinsic processes (56). The pattern of mutations in a sample, known as the mutational spectrum, will depend on the particular mutational and DNA repair processes involved (56). The chance of acquiring a specific single nucleotide substitution appears to depend (at least partially) on the adjacent nucleotides (56).

The probability of a particular somatic mutation appearing in a sequenced sample depends on both the rate at which the mutation occurs in cells and the strength of selection on the mutation once it has occurred. In genes where no mutations of any kind convey a growth advantage/disadvantage, the mutations detected by DNA sequencing will simply be an unbiased sample of the mutations produced by the spectrum. Mutations which convey a growth advantage to the cell are more likely to appear in multi-cellular clones that are detected in the sequencing data, and mutations that convey a disadvantage are less likely to be observed.

The model of neutral null hypothesis is created by combining the mutational spectrum for the dataset (Methods 5.8) and assigning an expected probability of occurrence to each mutation. Those probabilities are then assigned to the mutation scores (which could be $\Delta\Delta G$, distance from a substrate site, whether the mutation sits on an ion-binding site etc.) to construct an expected distribution of those scores (Figure S1). The observed distribution of scores can then be statistically compared (Supplementary Methods) to the expected distribution (Figure S1)."

(5) Hotspot mutations in cancer were discussed specifically for FBXW7, but not for NOTCH1. What is the reason for this?

We wanted to investigate how the observed mutations in normal skin and oesophagus might affect the binding of FBXW7 to its substrates. To directly test substrate binding using our method would require each mutation in the region to be given scores representing how they impact binding with NOTCH1 NICD (and potentially other substrates). This would have required a very large number of experiments or highly computationally expensive simulations, either of which would have been impractical for our study. Instead, we used the comparison with cancer hotspots, which have a known effect on FBXW7-NOTCH1 NICD interaction, to explore this question in an indirect manner.

In NOTCH1 we were able to assign a score for every mutation for each of the mutation categories we were investigating, and could directly statistically test selection of these mutation types. We therefore did not require a comparison with cancer hotspots to explain the mutational impacts under selection in NOTCH1 in the normal tissues.

(6) "The peak of missense mutations in the oesophagus mirrors that of the skin data (Figure S10a, Figure 4d),..." There is a mislabeling here. It should be Figure 4b instead.

We thank the reviewer for spotting this mistake. We have corrected the text.

(7) Please check Figure 4B. According to the description, there should be two figures, and it seems that the author missed one.

We thank the reviewer for spotting this mistake. This was an error in the figure caption. Figure 4b is only one graph. The second figure referred to in the erroneous caption (containing the equivalent graph for the oesophagus data) is Supplementary figure 10a.

We have corrected the caption.

REVIEWERS' COMMENTS:

Reviewer #1 (Remarks to the Author):

The authors have made a great effort to address the comments and to improve their manuscript. Comments have been answered properly.

Reviewer #2 (Remarks to the Author):

The authors have addressed all my questions.